# Isolation and Characterization of Prophylactic Antimalarial Agents from *Ochna kibbiensis* Leaves

**Amina J. Yusuf** [1,*] **, Musa I. Abdullahi** [1]**, Ibrahim Nasir** [1]**, Abdulmajeed Yunusa** [2]**, Celestina O. Alebiosu** [1] **and Abubakar A. Muhammad** [1]

[1] Department of Pharmaceutical and Medicinal Chemistry, Faculty of Pharmaceutical Sciences, Usmanu Danfodiyo University, Sokoto P.M.B. 2346, Nigeria

[2] Department of Pharmacology and Therapeutics, College of Health Sciences, Usmanu Danfodiyo University, Sokoto P.M.B. 2346, Nigeria

[*] Correspondence: amina.yusuf@udusok.edu.ng; Tel.: +234-8036386793

**Abstract:** *Ochna kibbiensis* (Family: Ochnaceae) has been employed in ethnomedicine for the treatment of malaria and inflammation, among others. The aim of this study was to isolate and characterize prophylactic antimalarial agents from the leaves of *O. kibbiensis* against *Plasmodium berghei*, in vivo and in silico. The median lethal dose ($LD_{50}$) of the methanol extract and its fractions (hexane, dichloromethane, ethylacetate and butanol) was determined according to Lorke's method while the antimalarial effect of the extract and its fractions was investigated according to the method described by Peters prophylactic test using Chloroquine-sensitive *Plasmodium berghei* (NK65). All the extract/fractions exhibited $LD_{50}$ values $\geq 5000$ mg/kg with the exception of the n-butanol fraction (1702.94 mg/kg), which indicate that the plant is non-toxic. Dichloromethane fraction exhibited a significant ($p < 0.05$) and dose-dependent prophylactic effect with 47.62, 85.12, and 100.0% prophylaxis (at 500, 250, and 125 mg/kg), while the least effect was observed by the butanol fraction with a percentage prophylaxis of 64.29 and 76.19, respectively; the standard drug, pyrimethamine, had 95.24% prophylaxis. Based on the results obtained, dichloromethane fraction of *O. kibbiensis* was subjected to chromatographic purification, which led to the isolation of a mixture of two compounds identified as stigmasterol and β-sitosterol by analysis of the NMR spectral data and comparison with existing literature; the compounds exhibited good binding affinities (−5.129 and −4.889 kcal/mol) against *pf*LDH and a favorable ADMET profile. In conclusion, the leaves of *O. kibbiensis* have demonstrated a significant prophylactic antimalarial activity and the two known steroids (stigmasterol and β-sitosterol) were isolated from the dichloromethane fraction for the first time.

**Keywords:** *Ochna kibbiensis*; antimalarial prophylaxis; in silico; ADMET





## 1. Introduction

Malaria is a life-threatening disease caused by parasites that are transmitted to people through the bites of infected female Anopheles mosquitoes [1]. According to the World Malaria Report, there were 241 million cases of malaria in 2020 with an estimated mortality of 627,000 compared to 227 million cases in 2019 leading to an increase of over 69,000 deaths which were due to disruptions during the COVID-19 pandemic and a recent change in WHO's methodology for calculating malaria mortality [1,2]. Four African countries accounted for just over half of all malaria deaths worldwide: Nigeria (31.9%), the Democratic Republic of the Congo (13.2%), the United Republic of Tanzania (4.1%) and Mozambique (3.8%) [1]. Children under 5 years of age accounted for about 80% of all malaria deaths in the African regions.

An estimated 50 million travelers visit malaria endemic areas annually and about 30,000 malaria cases in non-endemic industrialized countries are reported yearly. Imported malaria remains a public health problem associated with high fatality rates in European

countries, with the United Kingdom, France, Italy and Germany accounting for about 70% [3–5]. Malaria can be prevented via vaccine, personal protection and chemoprophylaxis. However, the malaria vaccine is not on the near horizon, despite the reports of new data [6,7]. Personal protection, albeit an important tool, is often not sufficient, thus chemoprophylaxis remains the principal means to prevent malaria [8]. Prevention of malaria in travelers to endemic areas has been confined and is fully dependent on chemoprophylaxis [5]. Although malaria chemoprophylaxis refers to all malaria species, it is important to note that there is a distinction between *falciparum* malaria prophylaxis and the prophylaxis of the relapsing malaria species (*vivax* and *ovale*); thus, *falciparum* prophylaxis use has been complicated due to emergence of drug resistant strains, and virtually high costs and adverse reactions to medications, and there are virtually no drugs available for *vivax* prophylaxis, except primaquine [5].

Traditionally, antimalarial drugs have been developed as agents for dual indications (treatment and prophylaxis). There are at least three prophylaxis strategies of administration that have been utilized; the most common strategy is the administration of casual or suppressive drugs at efficacious prophylaxis doses throughout the period of exposure to malaria which must be continuous [9], a post-exposure PART regime which is required to prevent subsequent relapse of *P. vivax* [10] and an alternative approach called 'fire and forget' prophylaxis, or 'pre-exposure prophylaxis', in which travelers are given a single dose or short course regime of a long half-life drug at a treatment dose that will protect them throughout the duration of exposure [11]. However, this approach is currently unproven in clinical practice and no drug for malaria prevention is adequate and effective in all respects [12]. Thus, there is a need to search and develop new prophylactic antimalarial drugs.

*Ochna kibbiensis* (OK) belonging to the Ochnaceae family is a shrub or small tree found in tropical Africa from Guinea to southern and northern Nigeria with brilliant red calyx in fruit; the plant has been used in ethnomedicine to treat and/or manage wound infections, pain, inflammation and malaria [3]; it is also used as a laxative, antiseptic, stimulant and febrifuge, among others [13]. We have reported the antimicrobial [14] and anti-proliferative [15] activities of the plant. Previous phytochemical investigations on the leaf of the plant revealed the isolation of ochnaflavone as one of the major bioactive constituents of the ethyl-acetate fraction [15]. In this paper, we report the prophylactic antimalarial properties of *Ochna kibbiensis* leaves, isolation and characterization of two known steroids, stigmasterol and β-sitosterol and their effect against *Plasmodium falciparum* Lactate Dehydrogenase (*pf*LDH) in silico.

## 2. Results

### 2.1. Acute Toxicity Studies

The median lethal dose of the methanol leaf extract of OK and its fractions were found to be safe based on the estimated $LD_{50}$ value of $\geq$5000 mg/kg, *i.p.* with the exception of BFL which had 1702.94 mg/kg (Table 1).

**Table 1.** Median lethal dose ($LD_{50}$) of extract/fractions of *O. kibbiensis*.

| Extract/Fraction | $LD_{50}$ Value (mg/kg) |
| --- | --- |
| Methanol | $\geq$5000.00 |
| Hexane | $\geq$5000.00 |
| Dichloromethane | $\geq$5000.00 |
| Ethylacetate | $\geq$5000.00 |
| Butanol | 1702.94 |

### 2.2. Antiplasmodial Activity of O. kibbiensis—Prophylactic Test

The crude extract and fractions of OK exerted a significant ($p < 0.05$) prophylactic effect against *P. berghei* (Table 2). The highest % prophylaxis (97.62%) was recorded by MLE at the highest dose (500 mg/kg), which subsequently reduced to 61.31% at the medium dose (250 mg/kg), and there was a slight increase (73.81%) at the lowest dose (125 mg/kg). A decrease in the level of parasitaemia was observed with the standard drug, pyrimethamine (25 mg/kg) at 95.24% prophylaxis.

**Table 2.** Prophylactic effect of *OK* against *P. berghei berghei* infection in mice.

|   | Treatment (mg/kg/day) | | Average Parasitaemia $\pm$ SEM | % Prophylaxis |
|---|---|---|---|---|
| 1 | NS 0.2 mL | | 8.40 $\pm$ 0.93 | - |
| 2 | MLE | 500 | 0.20$\pm$ 0.20 * | 97.62 |
|   | MLE | 250 | 3.25 $\pm$ 0.48 * | 61.31 |
|   | MLE | 125 | 2.20 $\pm$ 0.80 * | 73.81 |
| 3 | HFL | 500 | 2.20 $\pm$ 1.02 * | 73.81 |
|   | HFL | 250 | 5.40 $\pm$ 1.57 ** | 35.71 |
|   | HFL | 125 | 1.20 $\pm$ 0.58 * | 85.71 |
| 4 | DFL | 500 | 4.40 $\pm$ 1.63 *** | 47.62 |
|   | DFL | 250 | 1.25 $\pm$ 0.95 * | 85.12 |
|   | DFL | 125 | 0.00 $\pm$ 0.00 * | 100 |
| 5 | EFL | 500 | 1.75 $\pm$ 0.63 * | 79.17 |
|   | EFL | 250 | 1.75 $\pm$ 1.03 * | 79.17 |
|   | EFL | 125 | 0.80 $\pm$ 0.58 * | 90.48 |
| 6 | BFL | 500 | 3.00 $\pm$ 1.73 * | 64.29 |
|   | BFL | 250 | 2.00 $\pm$ 0.91 * | 76.19 |
|   | BFL | 125 | 2.00 $\pm$ 0.77 * | 76.19 |
| 7 | PM | 25 | 0.40 $\pm$ 0.24 | 95.24 |

Key: NS = Normal saline; MLE = Methanol leaf extract of OK; HFL = Hexane fraction of OK; DFL = Dichloromethane fraction of OK; EFL= Ethylacetate fraction of OK; BFL = Butanol fraction of OK; PM = Pyrimethamine; OK = *Ochna kibbiensis*. Values were expressed as Mean $\pm$ SEM ($n = 5$). Values of the group with * are statistically significantly ($p < 0.05$) different from NS treated. Values of the group with superscript ** are statistically significantly different from Pyrimethamine treated groups. Values of the group with superscript *** are statistically significantly different from NS and Pyrimethamine treated groups.

Of all the fractions tested, DFL was the most active while BFL was the least active. DFL exhibited a significant ($p < 0.05$) and dose-dependent effect with 47.62, 85.12 and 100% prophylaxis at 500, 250 and 125 mg/kg, respectively, while BFL recorded its highest % prophylaxis (76.19%) at the lowest (125 mg/kg) and medium (250 mg/kg) doses and reduction of the prophylactic effect (64.24%) at the highest dose (500 mg/kg). EFL exerted a similar effect to that of BFL; however, the highest (500 mg/kg) and the medium (250 mg/kg) doses of the fraction had % prophylaxis of 79.17% while the lowest dose (125 mg/kg) exhibited a higher prophylactic effect (90.48%) compared to the standard drug, pyrimethamine (95.24%). HFL had 85.71, 35.71 and 73.81% prophylaxis at the graded doses (500, 250 and 125 mg/kg) employed in this study (Table 2).

### 2.3. Phytochemical Screening of DFL of OK

Preliminary phytochemical screening of DFL revealed the presence of flavonoids, steroids and triterpenes (Table 3).

### 2.4. Characterization of K4

Compound **K4** (28.4 mg) was isolated as a white crystalline substance with an uncorrected melting point range between 136–137 °C, and it tested positive for Liebermann–Burchard's reagents; comparison of the NMR data of the compound with literature is summarized in Tables 4 and 5, respectively.

**Table 3.** Phytochemical constituents of DFL of OK.

| Constituents | Test | Observation | Inference |
|---|---|---|---|
| Alkaloids | Mayer's<br>Dragendorff's | White-yellowish ppt<br>Orange-brownish ppt | - |
| Carbohydrates | Molisch's<br>Fehling's | Reddish colored ring<br>Red precipitate | - |
| Flavonoids | Ferric chloride<br>NaOH | Green precipitate<br>Yellow colour | + |
| Saponins | Frothing | Froth persists for 15 mins | - |
| Tannins | Lead Sub-acetate | Cream ppt | - |
| Steroids/terpenes | Lieberman-Buchard's<br>Salkowski's | Brown ring at interface<br>Reddish color | + |
| Anthraquinones | Bontrager's | Pinkish color | - |

Key: + = present; - = absent.

**Table 4.** Comparison of spectral data of Stigmasterol with literature.

| Position | $^{13}$C-NMR K4 | $^{13}$C-NMR<br>Yusuf et al. [16] | $^1$H-NMR K4 | $^1$H-NMR<br>Yusuf et al. [16] | DEPT K4 |
|---|---|---|---|---|---|
| 1 | 37.27 | 37.26 | - | 1.85 | $CH_2$ |
| 2 | 31.67 | 31.67 | 1.44 | 1.46 | $CH_2$ |
| 3 | 71.83 | 71.81 | 3.45 | 3.52 | CH |
| 4 | 42.31 | 42.31 | 2.21 | 2.27 | $CH_2$ |
| 5 | 140.76 | 140.76 | - | - | C |
| 6 | 121.72 | 121.71 | 5.28 | 5.35 | CH |
| 7 | 31.93 | 31.90 | 1.92 | 1.96 | $CH_2$ |
| 8 | 31.88 | 31.90 | - | 1.48 | CH |
| 9 | 50.16 | 50.16 | 0.94 | 0.93 | CH |
| 10 | 36.51 | 36.51 | - | - | C |
| 11 | 21.09 | 21.21 | - | 1.49 | $CH_2$ |
| 12 | 39.7 | 39.68 | - | 1.16 | $CH_2$ |
| 13 | 42.23 | 42.22 | - | - | C |
| 14 | 56.88 | 56.87 | 1.09 | 1.05 | CH |
| 15 | 24.37 | 24.36 | 1.56 | 1.56 | $CH_2$ |
| 16 | 28.91 | 28.92 | 1.77 | 1.7 | $CH_2$ |
| 17 | 55.98 | 55.96 | - | 1.13 | CH |
| 18 | 12.05 | 12.05 | 0.74 | 0.69 | $CH_3$ |
| 19 | 21.2 | 21.08 | 1 | 1.03 | $CH_3$ |
| 20 | 40.48 | 40.49 | - | 2.02 | CH |
| 21 | 23.06 | 23.07 | 0.94 | 1.02 | $CH_3$ |
| 22 | 138.31 | 138.31 | 5.10 | 5.10 | CH |
| 23 | 129.3 | 129.28 | 5.05 | 5.03 | CH |
| 24 | 51.25 | 51.24 | - | 1.53 | CH |
| 25 | 29.2 | 29.15 | - | 1.65 | CH |
| 26 | 18.98 | 18.98 | 0.85 | 0.82 | $CH_3$ |
| 27 | 19.39 | 19.40 | 0.79 | 0.78 | $CH_3$ |
| 28 | 25.4 | 25.40 | - | 1.15 | $CH_2$ |
| 29 | 12.24 | 12.25 | 0.80 | 0.80 | $CH_3$ |

*2.5. In Silico Studies of Compound Stigmasterol and β-Sitosterol*

2.5.1. Molecular Docking Analysis

Stigmasterol and β-sitosterol from *O. kibbiensis* were screened against *pf*LDH, and the docking scores are indicated in Table 6. The binding energies for the best poses of the compounds range from −5.129 to −4.889 kcal/mol, while the co-crystalline ligand NADH had the highest binding affinity (−10.106 kcal/mol).

**Table 5.** Comparison of spectral data of β-Sitosterol with literature.

| Position | ¹³C-NMR K4 | ¹³C-NMR Abdulmalik et al. [? ] | ¹H-NMR K4 | ¹H-NMR Abdulmalik et al. [? ] | DEPT K4 |
|---|---|---|---|---|---|
| 1 | 37.11 | 37.26 | 1.09 | 1.08 | $CH_2$ |
| 2 | 31.68 | 31.66 | - | 1.51 | $CH_2$ |
| 3 | 71.83 | 71.82 | 3.45 | 3.52 | CH |
| 4 | 42.32 | 42.3 | 2.21 | 2.28 | $CH_2$ |
| 5 | 140.76 | 140.76 | - | - | C |
| 6 | 121.72 | 121.72 | 5.28 | 5.35 | CH |
| 7 | 31.92 | 31.91 | - | 1.99 | $CH_2$ |
| 8 | 23.09 | 23.07 | - | 1.25 | CH |
| 9 | 24.31 | 24.3 | - | 1.18 | CH |
| 10 | 36.51 | - | - | - | C |
| 11 | 21.09 | 21.08 | 1.44 | 1.46 | $CH_2$ |
| 12 | 39.8 | 39.78 | - | 1.16 | $CH_2$ |
| 13 | 42.32 | - | - | - | C |
| 14 | 56.79 | 56.77 | - | 0.98 | CH |
| 15 | 29.2 | 29.16 | 1.56 | 1.58 | $CH_2$ |
| 16 | 28.24 | 28.24 | 1.77 | 1.85 | $CH_2$ |
| 17 | 50.16 | 50.14 | - | 1.09 | CH |
| 18 | 11.98 | 11.98 | 0.74 | 0.69 | $CH_3$ |
| 19 | 11.86 | 11.86 | 1 | 1.03 | $CH_3$ |
| 20 | 37.27 | 37.25 | - | 1.35 | CH |
| 21 | 19.05 | 19.03 | 0.94 | 0.92 | $CH_3$ |
| 22 | 33.98 | 33.95 | - | 1.32 | $CH_2$ |
| 23 | 26.14 | 26.08 | - | 1.17 | $CH_2$ |
| 24 | 45.86 | 45.84 | 1.92 | 1.91 | CH |
| 25 | 29.7 | 29.69 | - | 1.65 | CH |
| 26 | 18.78 | 18.78 | 0.79 | 0.82 | $CH_3$ |
| 27 | 18.78 | 18.78 | 0.79 | 0.82 | $CH_3$ |
| 28 | 23.09 | 23.07 | - | 1.22 | $CH_2$ |
| 29 | 19.81 | 19.81 | 0.85 | 0.85 | $CH_3$ |

**Table 6.** Docking scores of Stigmasterol, β-sitosterol and some standard ligands against *pf*LDH.

| Compound | PubChem ID | Docking Score (in kcal/mol) |
|---|---|---|
| Stigmasterol | 5280794 | −5.129 |
| β-sitosterol | 222284 | −4.889 |
| NADH (Co-crystalline ligand) | 439153 | −10.106 |

### 2.5.2. Biological Interactions

The 2D- and 3D-view of the molecular interactions of the clinically important amino acid residues of *pf*LDH, with stigmasterol and β-sitosterol, and NADH, are shown in Table S1, Supplementary Materials, Figures 1 and 2, respectively. The co-crystallized ligand, NADH, formed a conventional hydrogen bond with Gly29, Met30, Ile31, Asp53, Asn140, Phe100 and His195 in addition to the other hydrophobic interactions with Gly27, Ser28, Gly32, Ile54, Val55, Met58, Thr97, Ala98, Gly99, Thr101, Ile123, Ile119, Val138, Thr139, Leu163, Leu167, Arg171, Pro246, Tyr247 and Pro250, while the compounds from OK (stigmasterol and β-sitosterol) interacted with Asp53 via hydrogen bonds in addition to other major hydrophobic interactions with Gly29, Met30, Ile31, Asp53, Asn140, Phe100 and His195, among others.

### 2.5.3. ADMET Analysis
Drug-Likeness, Oral Bioavailability, and Pharmacokinetic Properties

The results of the in silico ADMET screening and drug-likeness properties of stigmasterol and β-sitosterol from *O. kibbiensis* are indicated in Table 7; the molecular weight of the compounds range from 412.69 to 414.71 g/mol. Both compounds indicated one hydrogen bond acceptor and donor with molar refractivity of 132.75 and 133.23, respectively, and TPSA value of 20.23. The water solubility values of the compounds expressed in terms of LogSw were between −5.47 and −6.19, and the compounds were predicted to be poorly soluble. Consensus log *p*-values representing lipophilicity were between 6.97 and 7.19. Both compounds had one Lipinski and Egan violations, two Muegge and three Ghose violations, zero Veber's violations and a bioavailability score of 0.55. The compounds had low GI absorption, and none of them are BBB permeant or Pgp substrates; none of the

compounds are potential inhibitors of cytochrome P450 enzymes, though stigmasterol is a potential inhibitor of CYP2C9.

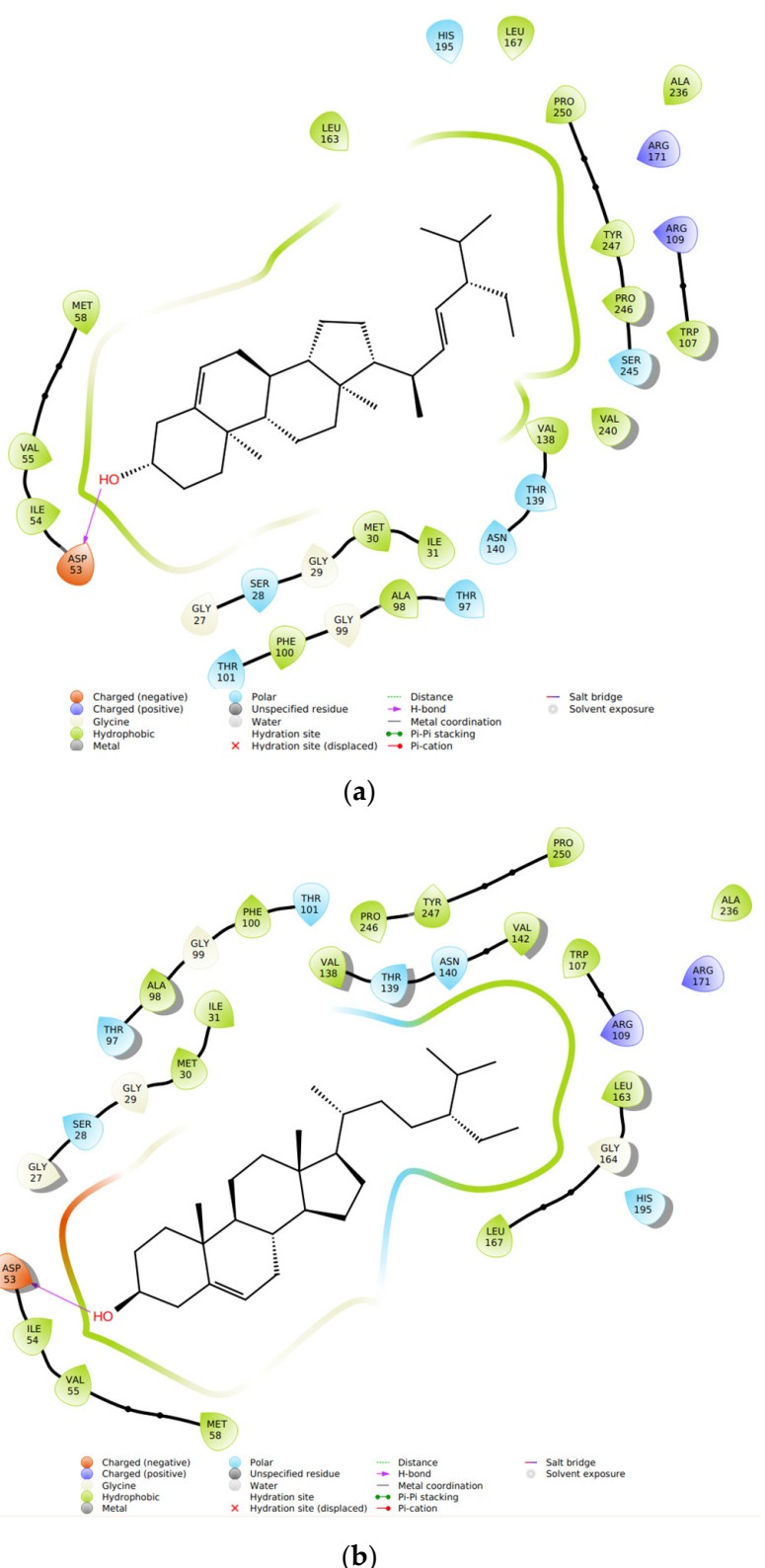

**Figure 1.** *Cont.*

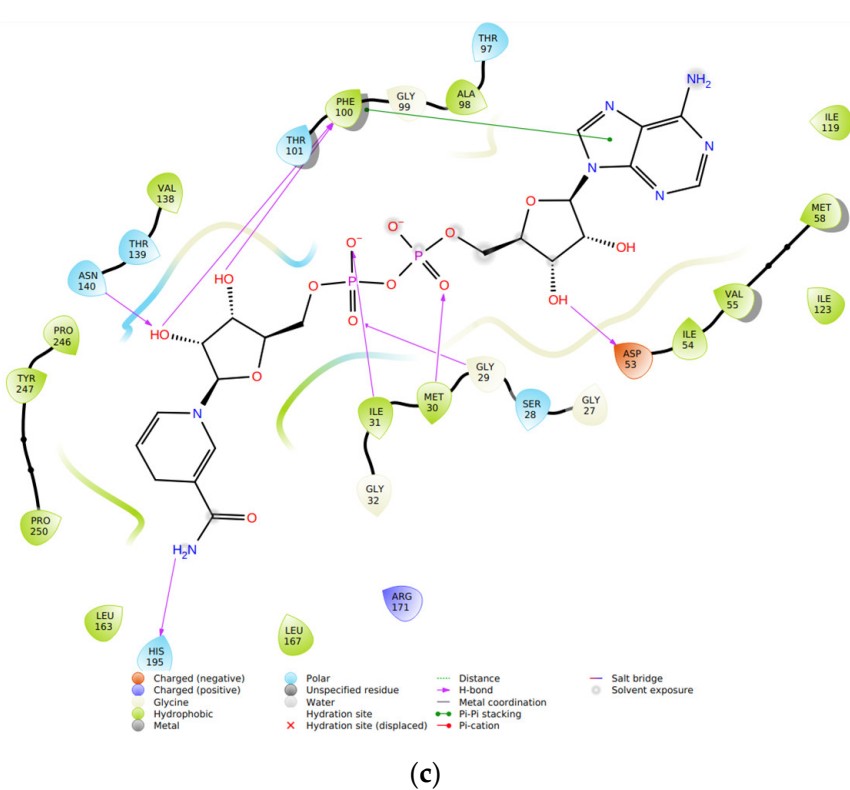

**(c)**

**Figure 1.** 2D view of the molecular interactions of (**a**) Stigmasterol; (**b**) β-sitosterol; (**c**) NADH with *pf*LDH.

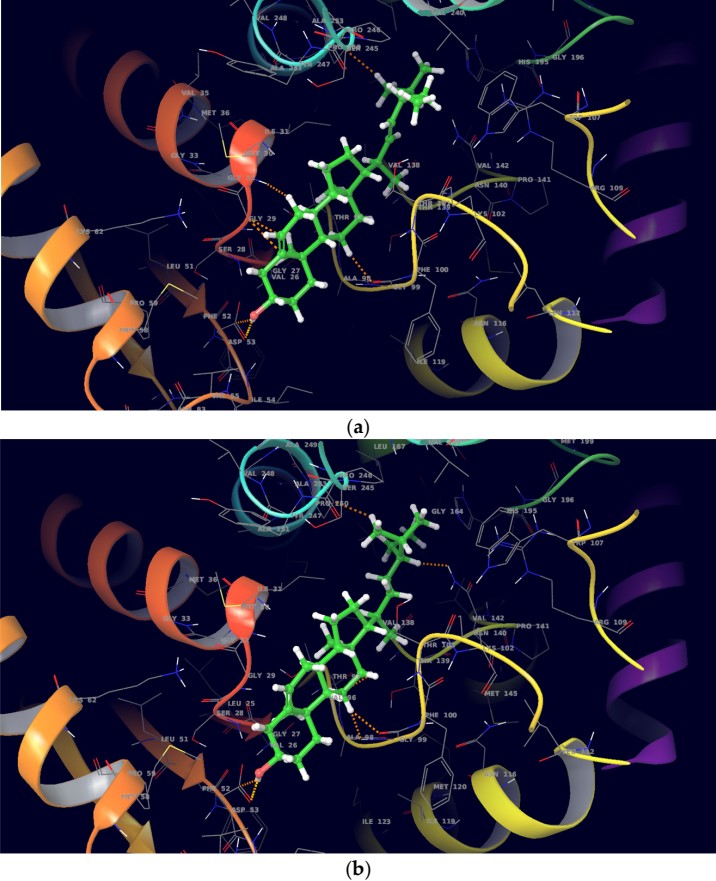

**Figure 2.** *Cont.*

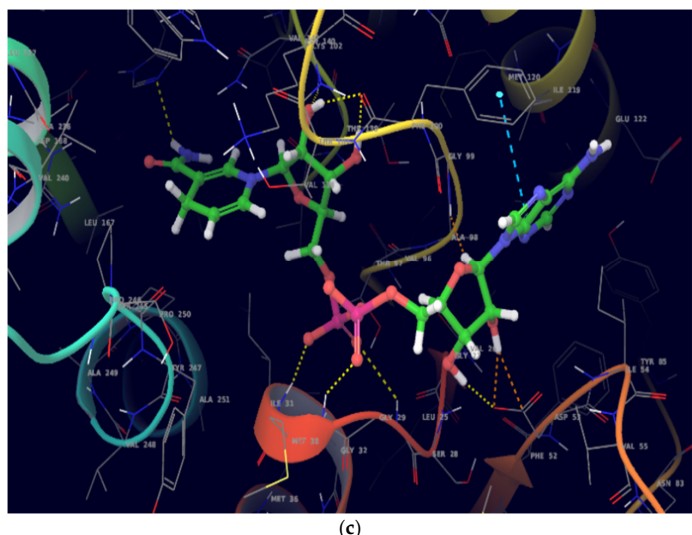

(**c**)

**Figure 2.** Three-dimensional view of the molecular interactions of (**a**) Stigmasterol; (**b**) β-sitosterol; (**c**) NADH with *pf*LDH.

**Table 7.** The drug-likeness, oral bioavailability, and pharmacokinetic properties of Stigmasterol and β-sitosterol from *O. kibbiensis*.

| Parameter | Stigmasterol | β-Sitosterol |
|---|---|---|
| Molecular weight (g/mol) | 412.69 | 414.71 |
| H-bond acceptors | 1 | 1 |
| H-bond donors | 1 | 1 |
| Molecular refractivity | 132.75 | 133.23 |
| TPSA ($\text{Å}^2$) | 20.23 | 20.23 |
| Silicos-IT LogSw | −5.47 | −6.19 |
| Silicos-IT class | Poorly soluble | Poorly soluble |
| Consensus Log P | 6.97 | 7.19 |
| Lipinski violations | 1 | 1 |
| Ghose violations | 3 | 3 |
| Veber's violations | 0 | 0 |
| Egan violations | 1 | 1 |
| Muegge violations | 2 | 2 |
| Bioavailability Score | 0.55 | 0.55 |
| log Kp (skin permeation) (cm/s) | −2.74 | −2.20 |
| GI absorption | Low | Low |
| BBB permeant | No | No |
| Pgp substrate | No | No |
| CYP1A2 inhibitor | No | No |
| CYP2C19 inhibitor | No | No |
| CYP2C9 inhibitor | Yes | No |
| CYP2D6 inhibitor | No | No |
| CYP3A4 inhibitor | No | No |

Toxicity Profile

The predicted $LD_{50}$ of Stigmasterol and β-sitosterol is 890 mg/kg, and they fall under the oral toxicity class 4; none of the compounds showed any indication of hepatotoxicity, carcinogenicity, mutagenicity, or cytotoxicity, but both compounds have a tendency of being immunotoxic (Table 8).

**Table 8.** Toxicity profile of Stigmasterol, β-sitosterol from *O. kibbiensis*.

| | Stigmasterol | β-Sitosterol |
|---|---|---|
| Predicted $LD_{50}$ (mg/kg) | 890 | 890 |
| Acute Oral Toxicity (c) | 4 | 4 |
| Hepatotoxicity | Inactive | Inactive |
| Carcinogenicity | Inactive | Inactive |
| Immunotoxicity | Active (* 99) | Active (* 99) |
| Mutagenicity | Inactive | Inactive |
| Cytotoxicity | Inactive | Inactive |

* Percentage probability.

## 3. Discussion

Considering the widespread usage of medicinal plants as remedies by a large proportion (>80%) of the world population as complementary and alternative medicines with little idea about their toxicity [18], it is very vital to conduct toxicity tests in order to evaluate the possible harmful side effects of these products. The increased use of these natural medicines is attributed to the fact that they are considered safe with little or no side effects compared to the orthodox medicines [19]. Thus, OK was subjected to an acute toxicity test to evaluate its median lethal dose ($LD_{50}$), and the findings demonstrated that the plant is relatively safe and acceptable. These values recorded for the methanol extract and its fractions agreed with the values reported for a related Ochna species (*O. schweinfurthiana*) at 5000 mg/kg orally [20].

The methanol leaf extract of OK and its fractions exhibited significant ($p < 0.05$) activity, which was comparable to pyrimethamine in the prophylaxis assay. DFL exhibited a very strong prophylactic antimalarial effect against *P. berghei*. In a study by Mil-Homens et al. [21], dichloromethane fraction was shown to exhibit the highest antimalarial activity compared to the other extracts and fractions studied. A decline in the prophylactic activity of MLE and HFL of OK at the medium dose (250 mg/kg) might be related to saturation at the active sites or the action of other endogenous substances in the body or activity on other body organs of the experimental animals [20]. Even more so, increasing the dose of BFL will lead to a decrease in prophylactic effect as clearly indicated in this study.

The extracts of OK may act by inhibiting the multiplication of the parasites as well as direct cytotoxic effect on the parasites [22] or by modulating the membrane properties of the erythrocytes there by preventing parasite invasion [23]. Pyrimethamine acts by inhibiting the dihydrofolate reductase of plasmodia, thereby blocking the biosynthesis of purines and pyrimidines, which are essential for DNA synthesis and cell multiplication, which leads to a failure of nuclear division at the time of schizont formation in erythrocytes and liver [24–27]. Generally, prophylactic antimalarials work by either disrupting the initial development of malaria parasites in the liver (casual activity), suppressing the emergent asexual blood stages of parasite (suppressive activity) or preventing the relapses induced by the latent liver forms (hypnozoites) of relapsing *P. vivax* and *P. ovale* malaria (presumptive antirelapse therapy) [10,27,28]. Thus, *O. kibbiensis* might act via the same mechanism. In addition, a higher prophylactic effect exhibited by DFL of OK might also be linked to the presence of secondary metabolites such as steroids, triterpenes and flavonoids that were present in the fraction [13].

Following the promising effects obtained in the prophylactic antimalarial effect using the repository test, the DFL of OK being the most active fraction was subjected to chromatographic studies which resulted in the isolation of a mixture of known steroids (**K4**). The isolated compound(s) appeared as a white crystalline substance with an uncorrected melting point ranging from 136–137 °C and gave a positive reaction when tested with Liebermann–Burchard's reagent, an indication of the presence of a steroid [16,29]. The [1]H-NMR spectrum of K4 (600 Hz, in CDCl$_3$) revealed the presence of three olefinic resonances at $\delta_H$ 5.28, 5.10 and 5.05, an oxy-methine proton at $\delta_H$ 3.45 and six methyl signals at $\delta_H$ 0.74, 1.00, 0.94, 0.85, 0.79 and 0.80, characteristic of a steroidal nucleus [16]. The [13]C-NMR and APT experiments of **K4** showed 29 carbon signals constituting four olefinic carbons at $\delta_C$ 140.76, 121.72, 138.31 and 129.28 corresponding to C-5, C-6, C-22 and C-23, respectively [30] and two angular methyl groups at $\delta_C$ 12.05 and 21.20 corresponding to C-18 and C-19, respectively [16,31]. The signal at $\delta_C$ 71.83 clearly indicated the presence of a β-hydroxyl group at C-3 typical of stigmasterol [16,30]. Additional peaks observed at $\delta_C$ 33.95 and 26.08 which were assigned to C-22 and C-23 indicated that **K4** is a mixture of stigmasterol and β-sitosterol. Based on the spectral data of compound **K4**, physicochemical tests and a direct comparison with existing data in the literature (Tables 3 and 4) [16,30–32], compound **K4** was elucidated and confirmed as a mixture of two compounds stigmasterol and β-sitosterol (Figure 3). Although the isolated compound(s) could not be tested for

prophylactic antimalarial activity due to insufficient quantity, the compounds have been reported to possess antimalarial activity [33–35]. Further bio-assay-guided isolation of the bioactive constituents responsible for the observed effect is presently ongoing in our lab.

**Figure 3.** Structures of Stigmasterol and β-Sitosterol.

On molecular docking analysis, stigmasterol and β-sitosterol showed lower docking scores against *pf*LDH compared to the co-crystallized ligand, NADH. Studies have shown that *pf*LDH is composed of NADH-binding sites with amino acid residues such as Gly-27, Ser-28, Gly-29, Phe-52, Asp-53, Ile-54, Thr-97, Ala-98, Gly-99, Phe-100, Thr-139 and Asn-140, and the substrate-binding site constituting Lys-198, Met-199, Val-200, Leu-201, Glu-226, Phe-229, Asp-230, Val-233, Lys-314 and Glu-317 [36]. In this study, stigmasterol and β-sitosterol interacted with clinically important amino acid residues similar to that of NADH; a similar study was reported by Read et al. [37].

ADMET analysis indicated stigmasterol and β-sitosterol to be poorly soluble with low GI absorption and none of the compounds are either BBB permeant or Pgp substrate. Both compounds had one Lipinski violation for drug-likeness. None of the compounds violate the general rule of five or Lipinski's rule for drug-likeness and both compounds are potential inhibitors of cytochrome P450 enzymes (CYP1A2, CYP2C19, CYP2D6 and CYP3A4); however, stigmasterol is a potential inhibitor of CYP2C9 and thus could lead to alteration of metabolism, which may lead to drug–drug interaction [38]. Moderate toxicity was recorded for both compounds based on the predicted $LD_{50}$ value 890 mg/kg and toxicity class of 4. However, both compounds have a tendency to be immunotoxic with a percentage probability of 99%.

## 4. Materials and Methods

### 4.1. Collection and Identification Plant Materials

The whole plant of OK was collected from Samaru Zaria, Northern Nigeria in the month of December, 2020. It was identified and authenticated by Mal. Namadi Sanusi at the Herbarium unit Department of Botany, Ahmadu Bello University Zaria, Nigeria by comparing with a voucher specimen number (573).

### 4.2. Extraction and Partitioning

The leaves of the plant were air dried, pulverized and preserved according to the methods described in the African Pharmacopoeia [39]. The pulverized plant material (3.73 kg) was extracted with 90% methanol (25 L) using a maceration method for 8 days, and the solvent was evaporated *in vacuo* using a rotary evaporator at 40 °C to yield a residue (825.51 g) referred to as the methanol leaf extract (MLE). The extract (770 g) obtained was suspended in distilled water to obtain water soluble and water insoluble portions; the water-soluble portion was then partitioned using solvents of increasing polarity to obtain

dichloromethane (DFL, 2.16 g), ethyl acetate (EFL, 34.69 g), butanol (BFL, 129.28 g) and residual aqueous fractions. The water insoluble portion (370.99 g) was washed with *n*-hexane to obtain a hexane (HFL, 5.59 g) fraction.

### *4.3. Antimalarial Studies*
### 4.3.1. Experimental Animals

Locally bred adult Swiss albino mice of either sex (19–30 g body weight) were acquired from the Animal House Facility of the Department of Pharmacology and Therapeutics, Ahmadu Bello University Zaria, Nigeria. The animals were fed with laboratory diet and water *ad libitum* and maintained under standard conditions in clean cages under normal 12 h light, 12 h dark cycles. All experimental procedures followed the ethical guidelines for the care and use of laboratory animals as provided by the Usmanu Danfodiyo University Research Policy and accepted internationally (UDUS/UREC/2020/001).

### 4.3.2. Malaria Parasite

A mouse-infective chloroquine-sensitive strain of *Plasmodium berghei* NK-65 was obtained from the National Institute of Medical Research, Lagos. The parasites were kept alive by continuous intra-peritoneal passage in mice.

### 4.3.3. Acute Toxicity Studies

The median lethal dose ($LD_{50}$) of MLE of OK, its n-hexane, dichloromethane, ethylacetate, and n-butanol fractions was determined using Lorke's method [40]. The study was divided into two phases. In phase one, nine (9) mice were divided into three groups containing three mice each. Groups 1, 2 and 3 received 10, 100 and 1000 mg/kg of each of the extract and fraction separately. Based on the results of the first phase, each treatment group received the following doses; MLE, EFL, DFL and HFL = 1600, 2900 and 5000 mg/kg; BFL = 600, 1000, 1600 and 2900 mg/kg. Animals in all groups were observed for any sign and symptoms of toxicity and mortality for 24 h, and the route of administration was intraperitoneal.

### 4.3.4. Antiplasmodial Screening
Parasite Induction

A standard inoculum of $1 \times 10^7$ of parasitized erythrocytes from a donor mouse in volume of 0.2 mL was used to infect the experimental animals intra-peritoneally.

Prophylactic Test—Residual Infection

The prophylactic activity of MLE of OK was tested according to the method described by Ryley and Peters [41]. Twenty-five (25) mice were randomly divided into 5 groups of 5 mice each. Groups 1, 2 and 3 were treated orally with 500, 250 and 125 mg/kg of MLE respectively, daily. Group 4 received 25 mg/kg of Pyrimethamine (positive control), while group 5 received 10 mL/kg of distilled water and served as the negative control. Treatments continued daily for four (4) consecutive days, and all the mice were infected with the parasite on the 5th day. Blood smears were then made from each mouse 72 h after inoculation/infection. Increase or decrease in parasitaemia was then determined microscopically. The average percentage prophylaxis was calculated as $100 \times [(P-Q)]/P$, where P is the average parasitaemia in the negative control group, and Q is the average parasitaemia in the test group [42]. The above procedure was repeated for the hexane, dichloromethane, ethylacetate and butanol fractions of *O. kibbiensis*. DFL was the most active fraction, and thus it was subjected to further fractionation using chromatographic techniques.

### *4.4. Chromatographic Fractionation of DFL*
### 4.4.1. General Experimental Procedures

Thin-layer chromatography (TLC) was carried out using silica gel 60 $GF_{254}$ precoated aluminum sheets (Sigma Aldrich, Germany). Column chromatography was conducted us-

ing LOBA Cheme silica gel (60–200) mesh. Spots on TLC plates were visualized by spraying with 10% $H_2SO_4$ followed by heating at 105 °C for 10 min. The melting point was determined on an Electro thermal melting point apparatus. NMR data were recorded on a Bruker AVANCE (III) spectrometer (600 MHz) with a residual solvent as the internal standard.

### 4.4.2. Phytochemical Screening of DFL

Fraction DFL was subjected to phytochemical tests to determine the presence of different secondary metabolites such as flavonoids, alkaloids, steroids and triterpenes, among others [13].

### 4.4.3. Isolation Procedures

DFL of the water-soluble portion of the MLE was loaded over a well packed silica gel column and eluted gradiently with the mixtures of *n*-hexane and ethyl acetate (9:1, 7:1, 5:1, 3:1 and 1:1) as mobile phase. Eluents were collected in 30 mL and were monitored using TLC. Nine major fractions (A to I) were obtained from the pooling of 120 collections made. Fractions C and D were subjected to purification using a silica gel column repeatedly, which afforded a white crystalline substance (14 mg) coded **K4**. Compound **K4** was subjected to TLC analysis using hexane: ethylacetate (6:1 and 5:1) as solvent systems, which revealed a single homogenous pink spot after spraying with 10% sulphuric acid and heated in an oven at 105 °C; the compound was also subjected to a Liebermann–Burchard's test, melting point determination and NMR analysis to elucidate its chemical structure.

### *4.5. Molecular Docking Studies*

#### 4.5.1. Protein Preparation

The Protein, *Plasmodium falciparum* lactate dehydrogenase (*pf*LDH), was prepared as previously described by Iwaloye et al. [43] and Johnson et al. [36]. The crystal structure of the protein obtained from the Protein Data Bank (PDB: 1T2C) repository was prepared in Glide (Schrödinger Suite 2021-2) using the protein preparation wizard panel. During the process, hydrogen was added, bond orders were assigned, disulfide bonds were created, and the missing side chains and loops were replaced with prime. Water molecules outside 3.0 Å of the heteroatoms were eliminated, and the protein structure was minimized and optimized using OPLS4 and PROPKA, respectively.

#### 4.5.2. Generation of Receptor Grid

The receptor grid was developed to define the location and size of the active site of the protein for ligand docking. This was accomplished with Schrödinger Maestro 12.8's receptor grid generation tool. The active site of the protein's co-crystalized ligand (1,4-dihydronicotinamide adenine dinucleotide) was employed as the scoring grid [43,44].

#### 4.5.3. Ligand Preparation

The crystal structures of stigmasterol and and β-sitosterol isolated from the leaves of *O. kibbiensis* were retrieved from Dr. Duke's Phytochemical and Ethnobotanical Databases, and the standard ligands (NADH) were prepared using the Ligprep panel of Maestro 12.8, Schrödinger Suite 2021-2, as previously reported [36,45]. Low-energy 3D structures with acceptable chiralities were generated. Each ligand's ionization state was formed at a physiological pH of 7.2 ± 0.2. Stereoisomers of each ligand were computed by keeping certain chiralities constant while varying others.

#### 4.5.4. Protein–Ligand Docking

The molecular docking analysis was performed on Schrödinger Suite 2021-2 utilizing the Glide–Ligand Docking panel of Maestro 12.8. The prepared ligands and the receptor grid file were loaded into Maestro's work space, and the ligands were docked into the protein's binding site. The vdW radius scaling factor was set to 0.80 for ligand atoms, with a partial charge cut-off of 0.15, and the flexible ligand sampling option was employed [45].

#### 4.5.5. ADMET Profiling

In silico predictive models were used to determine the ADMET properties of the test compounds i.e., stigmasterol and and β-sitosterol. The SwissADME server was used to assess the ADME properties of the compounds [46,47] while the ProTox-II online server was used to predict the toxicity profile of the compounds [48].

#### 4.6. Statistical Analysis

Results were expressed as mean ± standard error of mean (SEM). Statistical analysis of control and test data was based on simple one-way ANOVA, and Dunnett's post hoc test was used for different doses within a group. A Bonferroni post hoc test was used to compare results between the means at the significance level considered at $p < 0.05$.

### 5. Conclusions

The leaves of *O. kibbiensis* were found to be relatively safe based on the $LD_{50}$ values obtained, and the plant has demonstrated significant antimalarial activity with prophylactic effects at the graded doses employed in this study. In addition, a mixture of Stigmasterol and β-Sitosterol were isolated from the leaf of *O. kibbiensis*, and the compounds have shown good binding affinities against *Plasmodium falciparum* lactate dehydrogenase and a favorable pharmacokinetic profile in silico. Thus, the observed effect could be partly attributed to the presence of these compounds.

**Supplementary Materials:** The following supporting information can be downloaded at: https://www.mdpi.com/article/10.3390/ddc2010003/s1, Table S1: Molecular interactions of Stigmasterol, β-sitosterol, Chloroquine, Pyrimethamine and NADH with *pf*LDH.

**Author Contributions:** Conceptualization, A.J.Y.; methodology, A.J.Y., M.I.A., I.N. and A.Y.; formal analysis, A.Y.; investigation, A.J.Y., A.A.M. and C.O.A.; data curation, A.J.Y.; writing—original draft preparation, A.J.Y.; writing—review and editing, all authors; supervision, M.I.A. All authors have read and agreed to the published version of the manuscript.

**Funding:** This research work was funded by the Nigerian Tertiary Education Trust Fund (TETFund) Research Projects (RP) Intervention via the Institutional Based Research (IBR) Grant with Grant reference No. TETFUND/DR&D/CE/UNIV/SOKOTO/RP/VOL.1, Batch A, Serial no. 46.

**Institutional Review Board Statement:** The animal study protocol was approved by the Usmanu Danfodiyo University Research and Ethics Committee (UDUS/UREC/2020/002, 8 January 2022).

**Informed Consent Statement:** Not applicable.

**Data Availability Statement:** Not applicable.

**Acknowledgments:** We acknowledged the efforts of Yusuf Sani from the Department of Pharmacology and Therapeutics, College of Health Sciences, UDU Sokoto for his diligent support in conducting the antimalarial assay, Mal. Hamza Muhammad from the Department of Pharmaceutical and Medicinal Chemistry, UDU Sokoto for participating in the extraction protocols, and Vuyisa Mzozoyana, School of Chemistry and Physics, University of Kwa-Zulu Natal-Durban, South Africa, for running the NMR spectroscopy.

**Conflicts of Interest:** The authors declare no conflict of interest.

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
