# Peer review of "Isolation and Characterization of Prophylactic Antimalarial Agents from Ochna kibbiensis Leaves"

_ddc, doi:10.3390/ddc2010003_

Round 1

Reviewer 1 Report

In this paper, the authors have performed isolation and characterization of the extracts Ochna kibbiensis leaves and evaluated the prophylactic antimalarial activity. The work is clearly presented, however it seems two studies have been stitched together in the manuscript.

Specific comments:

1) It is not very clear how the present work is providing new findings from the previous report (Ibrahim et al, 2015; ref 18).

2) In the current work, the authors present prophylactic antimalarial activity of the isolated extracts. The parasite burden is only presented after 72h post-infection.  As the prophylaxis is never 100%, It is important to report survival of animal 28 days post-infection and mean survival time.

3) Regarding molecular docking studies,  the rationale of using PfLDH for docking of stigmasterol , sitosterol and standard drugs is not clear. PfLDH is not the target of chloroquine and pyrimethamine.

Author Response

Response to Reviewer 1 Comments

Point 1: It is not very clear how the present work is providing new findings from the previous report (Ibrahim et al, 2015; ref 18).

Response 1: Ibrahim et al., 2015, worked on the curative and suppresive antimalarial activity of a related Ochna species (i.e. Ochna schweinfurthiana) which is an entirely different work from ours. We comapred the median lethal dose of the plant they worked on to ours. Our work focus on Prophylactic effect of a different Ochna speciee i.e. Ochna kibbienses.

The comment have been addressed in the manuscript  

Point 2: In the current work, the authors present prophylactic antimalarial activity of the isolated extracts. The parasite burden is only presented after 72h post-infection.  As the prophylaxis is never 100%, It is important to report survival of animal 28 days post-infection and mean survival time.

Response 2: Our study was designed to check the prophylactic effect of Ochna kibbiensis 72 h post infection

Point 3: Regarding molecular docking studies, the rationale of using PfLDH for docking of stigmasterol, sitosterol and standard drugs is not clear. PfLDH is not the target of chloroquine and pyrimethamine.

Response 3: Plasmodium lactate dehydrogenase (pLDH), a terminal glycolytic enzyme involved in the bioconversion of pyruvate to lactate with a simultaneous oxidation of the cofactor NADH to NADþ (Samuel et al., 2021; Waingeh et al., 2013), is a well-established target for a novel antimalarial drug discovery (Shadrack et al., 2016). The pLDH is needed by the parasite to maximize glucose consumption for ATP production (required for biochemical processes including development, growth, and survival in the parasite) more than the host’s isoform, since the parasite lacks the tricarboxylic acid (TCA) cycle (Zakaria et al., 2020). Also, the amino acid sequence dissimilarity of the substrates specificity loop, and the kinetics of the pLDH when compared with the host’s isoform (hLDH) can avert off-target-related toxicity (Zakaria et al., 2020). Indeed, the development of small-molecule inhibitors against protein targets, like pLDH, is an effective therapeutic strategy against malaria.

Based on the above we choose that target for our study

Reviewer 2 Report

All the comments are presented in the manuscript. I suggest that the authors extend the introduction and link it better to the purpose of the manuscript. Maybe less information about the malaria and antimalarials in general and more focus on the actual topic of the manuscript, previous research on the two bioactive compounds, stigmasterol and beta-sitosterol. For example, this paper seems relevant to the manuscript, and it is not mentioned; https://doi.org/10.1186/s43088-021-00170-3. I strongly suggest that the authors pay more attention to the introduction.

Author Response

Response to Reviewer 2 Comments

Point 1: All the comments are presented in the manuscript. I suggest that the authors extend the introduction and link it better to the purpose of the manuscript. Maybe less information about the malaria and antimalarials in general and more focus on the actual topic of the manuscript, previous research on the two bioactive compounds, stigmasterol and beta-sitosterol. For example, this paper seems relevant to the manuscript, and it is not mentioned; https://doi.org/10.1186/s43088-021-00170-3. I strongly suggest that the authors pay more attention to the introduction.

Response 1: The research design/title is Isolation and Characterization of Prophylactic Antimalarial Agents from Ochna kibbiensis Leaves. It was not solely to isolate and characterize stigmatserol and beta-sitosterol from the plant rather, it was more of activity guided isolation. Focusing on the two compounds on the backgound section will be too specific and we wish to retain the background.

Reviewer 3 Report

The manuscript ID ddc-2012468, titled "Isolation and Characterization of Prophylactic Antimalarial Agents from Ochna kibbiensis Leaves" is exciting and details the isolation and characterization of natural compounds, as well as some biological assays in vivo and in silico. It will be of interest to researchers in the natural products area and also to the Drug and Drug Candidates journal readers. Some points need to be addressed or corrected before acceptance. The authors should carefully perform the plant identification by giving the authors' names. Moreover, the authors should explain why the species name is not accepted and is considered a Synonym of Ochna staudtii Gilg. Please eliminate the "n" in the n-hexane and n-butanol names because the IUPAC recommendations are against the use of "n". Additionally, it would improve the work and results if the authors give the chemical profile of the fractions and above all indicate the percentage of the isolated compounds in the dichloromethane extract. In that way the conclusion about the compounds contribution for the activity would be more accurate. 

Author Response

Response to Reviewer 3 Comments

Point 1: The manuscript ID ddc-2012468, titled "Isolation and Characterization of Prophylactic Antimalarial Agents from Ochna kibbiensis Leaves" is exciting and details the isolation and characterization of natural compounds, as well as some biological assays in vivo and in silico. It will be of interest to researchers in the natural products area and also to the Drug and Drug Candidates journal readers. Some points need to be addressed or corrected before acceptance. The authors should carefully perform the plant identification by giving the authors' names. Moreover, the authors should explain why the species name is not accepted and is considered a Synonym of Ochna staudtii Gilg. Please eliminate the "n" in the n-hexane and n-butanol names because the IUPAC recommendations are against the use of "n". Additionally, it would improve the work and results if the authors give the chemical profile of the fractions and above all indicate the percentage of the isolated compounds in the dichloromethane extract. In that way the conclusion about the compounds contribution for the activity would be more accurate. 

Response 1: All comments above have been addressed in the manuscript; n have been removed for n-hexane and n-butanol

Phytochemical profile of the most active fraction have been added

The mass of the compounds have been equally included, it was an oversight

Round 2

Reviewer 1 Report

Well the the response for point 2 and 3 is not satisfactory.

-As mentioned earlier, prophylaxis is not 100%. So the results reported for 3 days study may not be reflective of what is observed in standard 28 days test. Maybe authors can discuss this aspect in their discussion part.

-PfLDH is suggested to be a drug target, but it is not a target for chloroquine and pyrimethamine. So it is not clear how much reliable are docking results for PfLDH with CQ and PYR. Maybe that portion can be removed from the manuscript.

Author Response

Response to Reviewer 1 Comments

Point 1: Well the the response for point 2 and 3 is not satisfactory.

-As mentioned earlier, prophylaxis is not 100%. So the results reported for 3 days study may not be reflective of what is observed in standard 28 days test. Maybe authors can discuss this aspect in their discussion part.

Response 1: After 72 hours treatment, the animals were observed for 28 days no mortality was recorded and thus the average time of death cannot be captured. Thus, we followed a standard method

Point 1: -PfLDH is suggested to be a drug target, but it is not a target for chloroquine and pyrimethamine. So it is not clear how much reliable are docking results for PfLDH with CQ and PYR. Maybe that portion can be removed from the manuscript.

Response 1: This comment have been addressed as suggested by the reviewer
